# A Novel Exopolysaccharide Produced by *Sphingomonas* sp. MT01 and Its Potential Application in Enhanced Oil Recovery

**DOI:** 10.3390/polym17020186

**Published:** 2025-01-14

**Authors:** Mengting Lu, Xiaoxiao Lu, Weiyi Tao, Junzhang Lin, Caifeng Li, Shuang Li

**Affiliations:** 1College of Biotechnology and Pharmaceutical Engineering, Nanjing Tech University, Nanjing 211810, China; 202261118021@njtech.edu.cn (M.L.); luxiaoxiao@njtech.edu.cn (X.L.); 2College of Food Science and Light Industry, Nanjing Tech University, Nanjing 211810, China; taoweiyi@njtech.edu.cn; 3Research Institute of Petroleum Engineering and Technology, Shengli Oilfield Company, Sinopec, Dongying 257000, China; linjunzhang.slyt@sinopec.com (J.L.); licaifeng136.slyt@sinopec.com (C.L.)

**Keywords:** sphingan, biopolymer, exopolysaccharide, EOR, rheological property, MEOR

## Abstract

Sphingan is a crucial exopolysaccharide (EPS) produced by *Sphingomonas* genus bacteria with wide-ranging applications in fields such as food, medicine, and petroleum. In this study, a novel sphingan, named MT gum, was overproduced from the wild-type strain *Sphingomonas* sp. MT01 at a yield of 25.6 g/L in a 5 L fermenter for 52 h at 35 °C. The MT gum was mainly composed of D-glucose (65.91%) and L-guluronic acid (30.69%), as confirmed by RP-HPLC, with Mw 7.24 × 10^5^ Da. The MT gum exhibited excellent rheology and pseudoplasticity characteristics while maintaining function in high-temperature and high-salinity environments. The viscosity retention rates of MT gum (0.1%, *w*/*v*) were 54.06% (80 °C, 50,000 mg/L salinity) and 34.78% (90 °C, 50,000 mg/L salinity), respectively. The apparent viscosity of MT solutions (0.1%, *w*/*v*) was much higher than that of welan solutions under the same conditions. The MT gum also had the property of instant dissolution and completely swelled in 40 min. Meanwhile, the MT gum was resistant to 3–10 mg/L Fe^2+^ in the reservoir conditions, ensuring its application in offshore oil fields. These findings suggested that the biopolymer MT gum produced by the strain MT01 had significant potential in enhanced oil recovery (EOR) of high-temperature and high-salinity oil reservoirs.

## 1. Introduction

Exopolysaccharide (EPS) is a class of biopolymer produced by micro-organisms, exhibiting properties of thickening, emulsifying, and favorable rheology, with a wide range of applications in the food, pharmaceutical, and other industries due to their unique structure and properties [1]. Sphingan is a general term for EPS produced by *Sphingomonas* genus bacteria, such as gellan, welan, diutan, rhamsan, and sanxan. The sphingans are mostly species- or strain-specific heteropolysaccharides and are formed with a repeating tetrasaccharide backbone structure (D-glucose-D-glucuronic acid-D-glucose-L-rhamnose/D-mannose) with pyruvate, acetyl, and succinyl substitutions [2]. In contrast, the cellulose [3] consists of D-glucose, and the guar gum [4] consists of D-mannose and D-galactose, which are well-known biopolymers. Gellan gum has acetyl and glyceryl substituents but lacks glycosylated side chains and is used as a gelling agent within the food industry because of its exceptional gelling properties [5]. Welan gum contains L-rhamnosyl or L-mannosyl as a side chain, and its main chain has acyl and glyceryl substituents, with potential for application in the medicine and food industries due to its favorable rheological properties [1]. Sphingans have high viscosity and positive gelatinization properties, and their use in enhanced oil recovery (EOR) is a new field that has been developed in recent years [6].

EOR, often referred to as tertiary recovery, is a process employed to extract residual oil from a formation after the primary and secondary oil recovery phases [7]. Polymer flooding represents a significant oil recovery method in EOR and is widely used in both onshore and offshore oil reservoirs [7,8]. Especially in the Daqing oilfields in China, the recovery of blocks implementing industrial-scale polymer flooding has increased by over 12% [9]. As oil recovery progresses through the middle and late stages, the range of applications for polymer flooding has expanded from conventional reservoirs to double-high (high-temperature and high-salt) reservoirs [10]. Synthetic polymers, such as hydrolyzed polyacrylamide (HPAM), are commonly used as thickening agents during the flooding process. In environments characterized by elevated temperatures and salinity, the viscosity of HPAM undergoes a notable reduction, and high shear rates precipitate the degradation of the polymer [11]. The degradation of HPAM can generate toxic acrylamide monomers, which are potentially carcinogenic [12]. The viscosity of the displacing fluids is severely reduced under harsh conditions above 75 °C and 30,000 mg/L salinity in most oil reservoirs [13]. Lu et al. [14] reported that the viscosity of the HPAM under harsh conditions (90 °C, 200,000 mg/L salinity) was only 9.6% of the value under normal conditions (25 °C, pure water).

The biopolymers are environmentally friendly and have the potential to replace HPAM in polymer flooding. Compared with HPAM, the biopolymers exhibit a higher tolerance in terms of salinity, and additional chemicals can further improve their stability against salinity [12]. Discovering appropriate biopolymers for targeted oil reservoirs is significant for the polymer flooding of EOR [15]. In recent years, sphingans such as welan gum [16], diutan gum [10], and WL gum [17] have been considered to have the potential to be used in double-high reservoirs. In the context of double-high reservoirs, the main requirements for the biopolymers are thermal stability and salt stability [18]. The development of novel temperature- and salt-resistant biopolymers holds great significance in EOR regarding the oil recovery process entering the double-high reservoir.

In this study, an exopolysaccharide named MT gum was developed from an efficient producer, *Sphingomonas* sp. MT01. The monosaccharide composition and molecular weight of MT gum were analyzed. The temperature and salt resistance of the MT gum and the sticky fermentation broth (MT-Fer) were evaluated and compared with those of the commercially available welan gum. It indicated that MT gum was a novel sphingan and had potential application in enhanced oil recovery in double-high reservoirs, providing a new biopolymer product for EOR.

## 2. Materials and Methods

### 2.1. Identification of Strain MT01

The single colony of the wild-type strain MT01 isolated from the rhizosphere soil of tropical plants was cultured at 30 °C for 24 h in seed culture medium. A genomic DNA purification kit was used to extract the genomic DNA of strain MT01. The bacterial universal primers 27F (5′-AGAGTTTGATCCTGGCTCAG-3′) and 1492R (5′-GGTTACCTTGTTACGACTT-3′) were used to amplify the 16S rDNA gene. The PCR products were purified and sequenced by General Biol (Chuzhou, China). The results were analyzed using the BLAST program on the National Center for Biotechnology Information (NCBI) website (https://blast.ncbi.nlm.nih.gov, accessed on 6 August 2024). The phylogenetic tree was constructed using the neighbor-joining method of MEGA 11.

### 2.2. Strain, Media, and Culture Conditions

The freezing *Sphingomonas* sp. MT01 was activated at 30 °C for 24 h in seed medium containing 20 g/L sucrose, 1 g/L peptone FP400, 2.33 g/L (NH_4_)_2_HPO_4_, 1 g/L K_2_HPO_4_·3H_2_O, 0.1 g/L MgSO_4_·7H_2_O, and 0.003 g/L FeSO_4_·7H_2_O (initial pH 7.0–7.2). The fermentation medium for polysaccharide production contained 40 g/L carbon source, 1 g/L peptone FP400, 2.33 g/L (NH_4_)_2_HPO_4_, 1 g/L K_2_HPO_4_·3H_2_O, and 0.1 g/L MgSO_4_·7H_2_O (initial pH 7.0–7.2). The carbon sources were sucrose, xylose, and glucose, respectively. The fermentation temperature ranged from 30 to 40 °C. The commercially available peptone FP400 was produced by Angel Yeast Co., Ltd. (Yichang, China).

### 2.3. The Batch Fermentation of Strain MT01

Production of MT gum was performed by batch fermentation using a 5 L stirred autoclavable fermenter (Shanghai Chuyi Biotechnology Co., Ltd., Shanghai, China) containing 3.0 L of medium. The seed culture of strain MT01 (300 mL) was prepared in a flask in a rotary shaker at 30 °C and 200 rpm for 24 h. The initial temperature and pH were maintained at 30 °C and 7.0 (±0.1), respectively. The aeration rate and agitation speed were set at 0.5 vvm (air volume/culture volume/min) and 300 rpm, respectively, for the initial 12 h, then adjusted to 1.0 vvm and 400–1000 rpm to increase the dissolved oxygen concentration as the fermentation process went on.

### 2.4. Fermentation Profile Analysis

Cell growth was determined by measuring the optical density at 600 nm (OD_600_). The residual glucose content in the broth was measured using a biosensor equipped with a glucose oxidase electrode (SBA-40C, Shandong Academy of Sciences, Jinan, China). The apparent viscosity of fermentation broth was measured at 25 °C with a rotary viscometer (IKAROTAVISC lo-vi, Staufen, Germany). The suitable diluted broth was incubated at 70 °C for 30 min and then centrifuged at 8000 rpm for 20 min to separate the supernatant and cell pellet. The MT gum in the clear supernatant was obtained by acid precipitation (adjusting the pH of fermentation broth to 2–3 with 6 M HCl) [19], and the crude MT gum was dried at 60 °C to constant weight. The dry weight was measured, and the yield of MT gum was calculated as g/L in the broth. The dried MT gum samples were used for rheological properties measurements.

### 2.5. Monosaccharide Composition and Molecular Weight Analysis

#### 2.5.1. Preparation of Purified MT Polysaccharide

The crude MT gum was redissolved at a concentration of 1 g/L and the protein in the solution was removed using the Sevag method [20]. The solution containing MT polysaccharide was dialyzed against deionized water for 72 h; the purified MT polysaccharide was then freeze-dried to constant weight and then milled to obtain samples for further analysis.

#### 2.5.2. Monosaccharide Composition Analysis

Hydrolysis of MT polysaccharide: 8 mg of the sample was dissolved with 1 mL of 72% sulfuric acid and then immersed in water at 30 °C for 1 h. The hydrolysate was filled with water to 10 mL after being cooled and then filled with nitrogen. The mixture was sealed and kept in a 110 °C oil bath for 2 h. The 0.5 mL of treated mixture was cooled to room temperature, the pH was adjusted to neutral by NaOH, and then filled with deionized water to 1 mL.

The preparation of PMP derivatives of monosaccharides: The hydrolyzed sample was added with 0.2 mL of 0.3 M NaOH and 0.4 mL of methanol solution derived from 1-phenyl-3-methyl-5-pyrazolone (PMP). The mixture was kept in a water bath at 70 °C for 1 h after being filled with nitrogen. The mixture was added to 0.2 mL of 0.3 M HCl after being cooled to room temperature and filled with water to 2 mL. The obtained solution was shaken with 1.5 mL of chloroform, and the underlying chloroform was discarded after standing layering. After extraction three times, the sample was filtered by a 0.45 µm membrane and used for subsequent analysis and detection.

HPLC analysis: The monosaccharide of the PMP derivatives was analyzed using an Agilent 1200 HPLC system (Agilent Technologies, Santa Clara, CA, USA). The analysis column was a Zorbax Eclipse XDB-C18 (4.6 × 250 mm, 5 µm). The ultraviolet detection wavelength was 254 nm. The elution flow rate was 1.0 mL/min, and the temperature was 30 °C. The mobile phase A was 15% acetonitrile (diluted by 0.05 M K_2_HPO_4_, pH 6.8), and the mobile phase B was 40% acetonitrile (diluted by 0.05 M K_2_HPO_4_, pH 6.8) [21].

Sixteen kinds of monosaccharide standard products were prepared as standard liquor according to the above method (fucose, rhamnose, arabinose, galactose, glucose, xylose, mannose, fructose, ribose, galacturonic acid, glucuronic acid, aminogalactose hydrochloride, glucosamine hydrochloride, N-acetyl-D-glucosamine, guluronic acid, and mannuronic acid), and they were precisely configured as mixed standards. According to the absolute quantitative method, the mass of different monosaccharides was determined, and the molar ratio was calculated according to the molar mass of monosaccharides.

#### 2.5.3. Determination of Molecular Weight

The purified MT polysaccharide was dissolved at a concentration of 1 mg/mL with the mobile phase (0.1 M NaNO_3_, 0.01% NaN_3_). The solution was filtered through a 0.22 μm filter membrane. And then the sample was subjected to analysis using the Agilent 1260 Infinity II MDS system (Agilent Technologies, Santa Clara, CA, USA). The system was equipped with a refractive index detector and a double-angle laser light scattering detector. Chromatography was conducted on a PL aquagel-OH Mixed-H column (7.5 × 300 mm, 8 µm). The elution flow rate was 1.0 mL/min, and the temperature was 45 °C [22].

### 2.6. Determination of Dissolving Time

The MT gum obtained by acid precipitation was dissolved at a concentration of 5 g/L with water at room temperature (25 °C) and 65 °C, respectively, and maintained at pH 7.0–7.5 by adding NaOH solution. The apparent viscosity of the two samples was determined at 10, 20, 30, 40, 50, 60, 80, 100, and 120 min. The time at which the viscosity values began to stabilize was considered the dissolving time of the polymer, which was the minimum time required for the complete swelling of the polymer.

### 2.7. Rheological Property

Measurements of rheological properties: The MT gum obtained by acid precipitation was fully swelled in water at a concentration of 1.0% (*w*/*v*) and maintained at pH 7.0–7.5 by adding NaOH solution. The MT solution was further diluted with water to create solutions with concentration gradients of 0.2%, 0.4%, 0.6%, and 0.8% (*w*/*v*). The solutions were stirred on a magnetic stirrer for 1 h. The commercially available welan gum was produced by Xinhe Biochemical Co., Ltd. (Xingtai, China), and the welan gum solution was prepared at a concentration of 1.0% (*w*/*v*). The rheological properties were measured with a DHR-2 rheometer (TA Instruments, New Castle, DE, USA). The linear viscoelastic region was initially identified by conducting strain sweeps across a range of strains (0.01% to 100%) at a fixed frequency of 1.0 Hz. Then, the stain was fixed at 0.1% in order to ensure that both the energy storage modulus and loss modulus were within the linear viscoelastic region. Frequency sweeps were performed in the range of 0.01–1 Hz at 25 °C in oscillatory mode. Shear thinning real experiments were conducted in rate-controlled mode with shear rate measurements in the range of 0–100 L/s.

### 2.8. The Resistance of Temperature, Salinity, and pH

The stability of the MT gum to high temperatures and salinity: The thick fermentation broth of MT (MT-Fer) was directly diluted 5×, 10×, and 20× with tap water or brine water, respectively, and the diluted samples were incubated at 40 °C, 60 °C, 80 °C, 85 °C, 90 °C, 95 °C, and 100 °C for 10 h. Subsequently, the apparent viscosity of the treated samples was measured at the corresponding temperature with a rotary viscometer (IKAROTAVISC lo-vi, Staufen, Germany). The welan gum solution (0.1%, *w*/*v*) was used as the control. The brine water containing total salinity of 10,000 mg/L, 30,000 mg/L, and 50,000 mg/L was described, as shown in Table 1.

The stability of the MT gum to pH: The pH of MT-Fer diluted 5×, 10×, and 20× with water was adjusted to 4, 6, 8, 10, and 12 with 2 M HCl and 2 M NaOH. After 2–3 h of reaction, the apparent viscosity of the above solutions was measured with a rotary viscometer.

### 2.9. Stability at Shear Stress

In order to study the stability of MT gum under shear stress under the simulated double-high reservoir conditions, the experiments were conducted according to the method provided in [23]. The MT gum (0.1%, *w*/*v*) and welan gum (0.1%, *w*/*v*) dissolved in brine water (50,000 mg/L salinity) were exposed to a constant shear rate of 7.34 s^−1^ at 80 °C for 2 h, and the viscosity was measured every 30 s. To prevent the evaporation of the sample, glycerin was applied to cover the surface during testing.

### 2.10. Ferrous Ion Tolerance

The influence of Fe^2+^ (0–10 mg/L) on the viscosity of MT gum and welan gum was assessed with the method in [17]. To prevent the oxidation of Fe^2+^, 40 mM HCl was incorporated into the FeSO_4_ solution. To simulate the double-high environment, the MT gum (0.1%, *w*/*v*) and welan gum (0.1%, *w*/*v*) were dissolved in the brine water (50,000 mg/L salinity) with or without FeSO_4_ solution. All solutions were incubated at 80 °C for 10 h; the apparent viscosity of solutions was measured by a rotary viscometer.

## 3. Results

### 3.1. Molecular Identification of Sphingomonas sp. MT01

The colonial morphology of MT01 exhibited a yellow, smooth surface and secreted a large amount of sticky substance when grown on sugar plates (Figure 1a). The 16S rDNA sequence of strain MT01 (GenBank accession NO. SRR30104875) showed the highest similarity with *Sphingomonas* sp. ZKA42 by BLAST alignment analysis (Figure 1b). Therefore, the strain was named *Sphingomonas* sp. MT01 and deposited in the China Center for Type Culture Collection (CCTCC NO:M20232420).

### 3.2. Effects of Different Carbon Sources on Production

The production of polysaccharide by *Sphingomonas* species typically used sucrose or glucose as the carbon source. Figure 2 shows the effects of different carbon sources on the production of MT gum. *Sphingomonas* sp. MT01 was able to efficiently produce MT gum using sucrose, xylose, and glucose, and the apparent viscosity of fermentation broth was 4820 ± 40 mPa·s, 4910 ± 30 mPa·s, and 4800 ± 50 mPa·s, respectively. The polysaccharide yields of all three carbon sources were very close, at around 25–27 g/L. The ability of the MT01 strain to use xylose for polysaccharide production provided the feasibility for using lignocellulosic biomass. Considering that glucose was a relatively cost-effective carbon source, the glucose was finally selected as the carbon source.

### 3.3. Effects of Temperature on MT Gum Production

Usually, the fermentation temperature of the biopolymer produced by bacteria was 30 °C. However, a higher temperature was beneficial for fermentation process control. Table 2 shows the growth and production of *Sphingomonas* sp. MT01 under 30–40 °C. Strain MT01 was capable of undergoing fermentation at 30–37 °C. After 52 h of fermentation at 35 °C, the apparent viscosity reached 4800 ± 30 mPa·s, with a yield of 25.6 g/L. The synthesis of MT gum was severely inhibited at 40 °C. The biomass could reach 3.06 (OD_600_), but there was no polysaccharide accumulation.

Figure 3 shows the fermentation process under the optimal production conditions. The polysaccharide MT gum began to accumulate rapidly after 24 h of fermentation, and the apparent viscosity of the fermentation broth increased rapidly; as the glucose nearly exhausted at 52 h of fermentation, the fermentation could end.

### 3.4. Monosaccharide Composition and Molecular Weight of MT Gum

The results of monosaccharide composition and molecular weight are shown in Table 3. In general, the MT gum was an acidic polysaccharide and mainly consisted of D-glucose and L-guluronic acid, and the molar ratio was 65.91% and 30.69%, respectively.

The weight-average molecular weight values were 7.24 × 10^5^ Da, and the number-average molecular weight values were 2.70 × 10^5^ Da. The PDI (Mw/Mn) was 2.68, which indicated that the molecular weight distribution of MT gum was relatively uniform.

### 3.5. Solubility at Different Temperatures

The limited space available on offshore oil platforms makes it challenging to allocate significant quantities of polymers. It is therefore of great significance to study biopolymers with rapid dissolution properties. The MT gum was redissolved at a concentration of 5 g/L with water at room temperature (25 °C) and 65 °C; the dissolution curve is shown in Figure 4. The dissolution equilibrium of 5 g/L MT gum could be reached within a period of 40–50 min at 25 °C and 30–40 min at 65 °C. This indicated that MT gum had instant solubility properties and could complete full swelling within 1 h, which could be effective in reducing the time and space requirements on offshore oil platforms. These results are beneficial for batching on offshore oil production platforms.

### 3.6. Rheological Property of MT Gum

The steady-state shearing curves of MT gum solutions and welan solutions at different shear rates are shown in Figure 5a. All the solutions exhibited the typical properties of pseudoplastic fluids with shear-thinning behavior. The apparent viscosity of the solutions decreased with increasing shear rate. The phenomenon of shear thinning is typically attributed to the behavior of polysaccharide molecules undergoing stretching, deformation, or dispersion due to changes in shear force. Under low shear rates, polysaccharide molecules existed as aggregates, while at high shear rates, the aggregates dissociated gradually by the shear rate force, and individual molecules rearranged along the flow direction [24]. The apparent viscosity decreased as a result.

The linear dynamic viscoelastic properties of all solutions with different concentrations were quantified (as shown in Figure 5b). The storage modulus (G′) and loss modulus (G″) are parallel to a certain shear zone, which means the linear viscoelastic region [25]. The MT gum, as well as the welan gum, existed in the linear viscoelastic region of the measurement range, with G′ consistently exceeding G″, which indicated that the elasticity of the MT gum was superior to the viscosity in the linear viscoelastic region. With the measurement scope, the G′ and G″ of all solutions exhibited intersection points, suggesting that the solution underwent a transition from the solid state to the liquid state.

The fixed strain in the linear viscoelastic region was selected for frequency sweep. Figure 5c shows the correlation between G′, G″, and the oscillation frequency of the MT solutions. Under the condition that the G′ of a polymer solution is higher than G″, the elastic properties of the solution dominate, and a gel-like structure may be formed [25,26]. The MT gum solution at low concentration (0.2%, *w*/*v*) could undergo a phase transition from liquid-like to solid-like, and this intersection is the gel point. When the concentration of the solution was higher than 0.2%, the G′ of the MT gum and welan gum solutions was always higher than G″ in the range measured, which indicated that the gel-like structure of all solutions was stable. At the same concentration, the G′ and G″ of the MT gum solution were higher than those of the welan gum, and the value of G′ and G″ was closer than that of the welan gum at the same frequency. These results indicated that the internal network structures of MT gum were more stable [27]. In polymer flooding for EOR, the viscoelasticity of the flooding fluid affects the recovery. The viscoelastic polymer solution has pushing, drawing, and dragging effects on residual oil to improve its migration [17,28].

### 3.7. Temperature, Salt, and pH Tolerance

#### 3.7.1. Temperature Tolerance

The polysaccharide entanglement is primarily based on hydrogen bonding and van der Waals force interactions. The hydrogen bond between the main chains and the side chains, as well as the intramolecular van der Waals force, can interact to form a network structure [29]. High temperatures can weaken this interaction, leading to a decrease in viscosity. Biopolymers usually precipitate, or their molecular structures are destructed at temperatures higher than 80 °C, and thus the viscosity of their solutions begins to decrease [17,30]. The effects of temperature on the viscosity of MT gum are shown in Figure 6a. The viscosity of MT fermentation solutions (MT-Fer) diluted five times was 380 ± 3 mPa·s at 80 °C, and the viscosity retention rate was 61.60%. Therefore, we further reduced the concentration of polysaccharides for the temperature resistance experiment. The 10× and 20× diluted MT-Fer solutions also exhibited great stability at temperatures ranging from 20 °C to 60 °C; however, they lost viscosity at 80 °C.

#### 3.7.2. Salt Tolerance

High salinity also affects the viscoelasticity of the polysaccharides. The decrease in the apparent viscosity of the polymer solution is caused by the attraction of salt ions to the ions on the polymer molecules [31]. Inorganic ions with strong hydration also make the hydration layer around the polysaccharide molecules tighter, thus reducing the apparent viscosity [32]. For different dilutions of MT-Fer (5×, 10×, 20×), the addition of salinity had no significant effect on apparent viscosity (Figure 6b). After the salinity treatment, the viscosity retention rates of the MT-Fer solutions were above 85% in all cases.

#### 3.7.3. pH Tolerance

As shown in Figure 6c, the MT-Fer solutions were relatively stable in the pH 4–12 range. The viscosity of MT-Fer solutions slightly increased with the increase in pH. With the pH increased from 4 to 12, the viscosity of MT-Fer solutions (5×) increased from 559.9 ± 5 mPa·s to 583.9 ± 6 mPa·s. The increase in pH resulted in an increase in the carboxyl ionization of the polymer [33]. The ionization of the carboxyl group led to the extension of the molecule, thereby increasing the apparent viscosity [34]. However, the polysaccharide in MT-Fer could be precipitated at pH 2–3, resulting in the loss of viscosity. Among sphingan, Sanxan [35] also has the property of acid precipitation and can be used for biopolymer extraction. However, the mechanism of acid precipitation remains unclear.

#### 3.7.4. Temperature-Salinity Composite Experiments

Considering the practical applications in double-high reservoirs, temperature-salinity composite experiments were carried out in the study. The apparent viscosity of solutions is shown in Figure 7. The most striking feature was that the viscosity of MT solutions significantly increased at 80 °C after the addition of salinity. The viscosity of the MT-Fer solution (20×) was 50.5 ± 0.35 mPa·s (10,000 mg/L salinity), 48.1 ± 0.45 mPa·s (30,000 mg/L salinity), and 49.0 ± 0.2 mPa·s (50,000 mg/L salinity) at 80 °C. The apparent viscosity of the MT gum solution (0.1%, *w*/*v*) was 38.1 ± 0.2 mPa·s (10,000 mg/L salinity), 37.0 ± 0.1 mPa·s (30,000 mg/L salinity), and 37.9 ± 0.2 mPa·s (50,000 mg/L salinity) at 80 °C. The viscosity retention rates of MT samples treated with 50,000 mg/L salinity at 80 °C were 56.06% (20×, MT-Fer) and 51.63% (0.1%, *w*/*v*), respectively. However, the apparent viscosity of welan gum solution (0.1%, *w*/*v*) was only 23.1 mPa·s and the viscosity retention was 59.23% in the same conditions. MT gum exhibited higher apparent viscosity than welan gum at the same condition. Subsequently, the experiments were further carried out at 85 °C, 90 °C, 95 °C, and 100 °C, respectively. The results indicated that MT gum could still maintain a higher viscosity than welan gum at 90 °C. However, when the temperature rose to 95 °C and above, the MT gum lost its viscosity. This indicated that MT gum is a potentially appropriate material for EOR in double-high reservoirs.

### 3.8. Stability of MT Gum at Shear Stress

In order to study the viscosity variation of MT gum under the simulated reservoir conditions (50,000 mg/L salinity, 80 °C), the thixotropic test was conducted at a constant shear rate (7.34 s^−1^) for 2 h. The change in apparent viscosity over time is presented in Figure 8. After two hours of shear, the viscosity of the MT gum solution did not decrease significantly and remained at 41.8–34.2 mPa·s. The situation was similar for the control welan gum solution, although its viscosity was only 32.3–20.6 mPa·s.

### 3.9. Ferrous Ion Effects on the Viscosity

The reservoir formation water of offshore oil fields in China always contains higher concentrations of Fe2+ than onshore oil fields, and it is usually necessary to add iron remover or iron stabilizer to the water [36]. The presence of Fe^2+^ can significantly affect the viscosity of HPAM [37], thereby influencing the efficiency with which oil is displaced. The effects of Fe^2+^ on the viscosity of biopolymer solutions have been rarely reported. Figure 9a shows that 3 mg/L Fe^2+^ had no effect on the viscosity of all solutions under the simulated conditions. The viscosity retention of MT gum and welan gum was 107.14% and 103.13%, respectively. Therefore, the concentration of Fe^2+^ was further increased to 10 mg/L; the apparent viscosity of all solutions remained stable (shown in Figure 9b). These findings indicated that both MT gum and welan gum could resist Fe^2+^.

## 4. Discussion

The techniques used in traditional EOR often have some impact on the environment. Therefore, researchers are working to develop environmentally friendly alternative strategies, of which MEOR is a promising approach [38]. MEOR is a biotechnology developed in the middle and late stages of EOR. The principle of MEOR is to use microorganisms and their metabolites to enhance the mobility of crude oil and ultimately improve oil recovery [39]. The high-temperature and high-salt environment of double-high reservoirs inhibits the growth of microorganisms and could significantly affect the performance of the MEOR process [40,41]. Biosurfactants and biopolymers are the key functional metabolites beneficial for MEOR processes; the large-scale production of these products by ground fermentation can effectively overcome the above shortcomings. Commercially available biopolymers, such as xanthan gum [42], welan gum, and diutan gum, are considered to be suitable for polymer flooding in double-high reservoirs but have not yet been applied to large-scale fields. Perhaps the high cost of biopolymers is the main factor hindering their application. It is of great value to develop new thermo-salt-tolerant biopolymers for the enhanced oil recovery of double-high reservoirs.

The production cost of biopolymers is affected by factors such as raw materials, yield, fermentation time, and fermentation temperature. The reported optimal fermentation temperature for *Sphingomonas* species is around 30 °C, and their production time is more than 60 h (Table 4). *Sphingomonas* species always exhibit a poor capacity to tolerate stressful environments. Consequently, a large amount of cooling water is used in the fermentation process, particularly during summer production, resulting in high energy consumption [43]. Furthermore, long fermentation times will lead to an escalation in production costs, which will limit the application and large-scale production of sphingans. Liu et al. [44] expressed the global transcriptional regulator IrrE in *Sphingomonas* sp. NX-3 for welan gum production to improve the environmental tolerance of the strain. Liu et al. [45] obtained a mutant *Sphingomonas* sp. HT-1 through genetic engineering, which could grow at 37 °C. However, in this work, the strain *Sphingomonas* sp. MT01 had significant advantages due to its high fermentation temperature (35 °C, 37 °C), high yield, and short fermentation time. MT gum had the potential for large-scale production.

Sphingans are the general term for a class of exopolysaccharides produced by *Sphingomonas* species. Currently, the monosaccharide composition of the commercially available sphingans, including welan gum, gellan gum, and so on, mainly consists of D-glucose, D-mannose, D-glucuronic, and L-rhamnose (Table 5). However, the monosaccharide composition of MT gum mainly contains D-glucose and L-guluronic acid, and the molar ratio is 65.91% and 30.69%, respectively. In contrast to other sphingans, the monosaccharide component of the MT gum does not contain rhamnose, indicating that the MT gum is a novel sphingan.

Polymer solutions easily lose their viscosity in a double-high environment. This defect becomes the main limiting factor for polymer displacement in double-high reservoirs. So far, in China, chemical flooding for EOR has been applied to reservoirs with a formation temperature of 85 °C, a formation water salinity of 30,000 mg/L, a calcium and magnesium ion concentration of 1500 mg/L, and a formation oil viscosity of 1000 mPa·s [52]. Therefore, when evaluating whether polymers are suitable for double-high reservoirs, temperatures above 80 °C and salinity above 30,000 mg/L are usually used as starting points. The MT gum exhibited instant solubility, pseudoplasticity, and excellent rheological properties. The full swelling of MT gum could be completed in 40–50 min at room temperature (25 °C) and in 30–40 min at 65 °C, which helped to improve the efficiency of the use of oilfield facilities. The MT gum also exhibited high viscosity and stability at the same concentrations (0.5%, *w*/*v*) as other biopolymers, such as XP1 [15], HS-EPS [25], and WL [17]. Especially, the MT gum was able to work at concentrations as low as 0.1% (*w*/*v*) under conditions of 90 °C and 50,000 mg/L salinity. The MT gum also exhibited its resistance to 3–10 mg/L Fe^2+^ at low concentrations (0.1%), which suggested it was appropriate for polymer flooding in Chinese offshore oilfields. In EOR, it is necessary to ensure the stability of biopolymers under reservoir conditions during the time of application. The MT gum was not affected by prolonged exposure to shear stress (7.34 s^−1^) under the conditions of 80 °C and 50,000 mg/L salinity, which indicated that MT gum could be applied to double-high reservoirs. Additionally, the most imperative property of the polymer solution was its ability to generate viscosity at a minimum concentration. The apparent viscosity of MT gum solutions was much higher than that of the welan gum solutions under the same conditions (≤90 °C, 50,000 mg/L salinity). This means that the amount of MT gum in polymer flooding can be significantly reduced, and thus the cost can be significantly reduced. It is well known that welan gum demonstrates excellent viscosity retention under conditions of high temperature and salinity and has been used in oil production, which also means that MT gum has excellent prospects for EOR applications in double-high reservoirs.

## 5. Conclusions

In this work, strain *Sphingomonas* sp. MT01 could efficiently produce polysaccharide MT gum at 30–37 °C. The polysaccharide MT was a novel sphingan, mainly containing D-glucose (65.91%) and L-guluronic acid (30.69%). The MT gum demonstrated excellent thickening and high-temperature salt tolerance, and the viscosity retention rates of the MT gum (0.1%, *w*/*v*) were 54.06% (80 °C, 50,000 mg/L salinity) and 34.78% (90 °C, 50,000 mg/L salinity), respectively. The apparent viscosity of MT solutions was much higher than that of the welan gum solutions under the same conditions (≤90 °C, 50,000 mg/L salinity). The MT gum was resistant to Fe^2+^ (3–10 mg/L) and shear stress. Biopolymer MT gum had the potential to be used in double-high reservoirs.

## Figures and Tables

**Figure 1 polymers-17-00186-f001:**
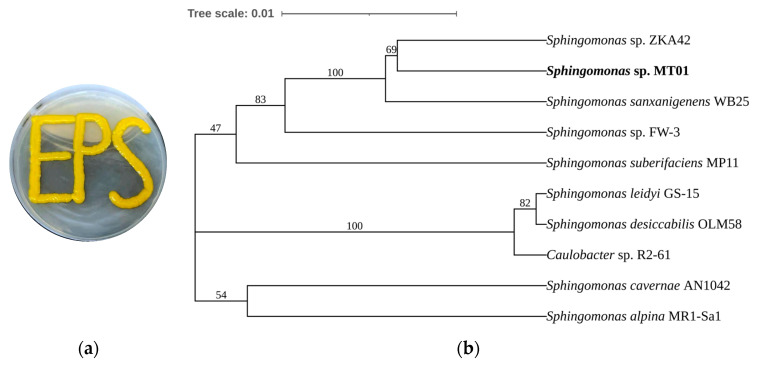
Morphological characteristics and identification of *Sphingomonas* sp. MT01. (**a**) The colony morphology of MT01. (**b**) Phylogenetic tree of MT01 based on 16S rDNA sequence.

**Figure 2 polymers-17-00186-f002:**
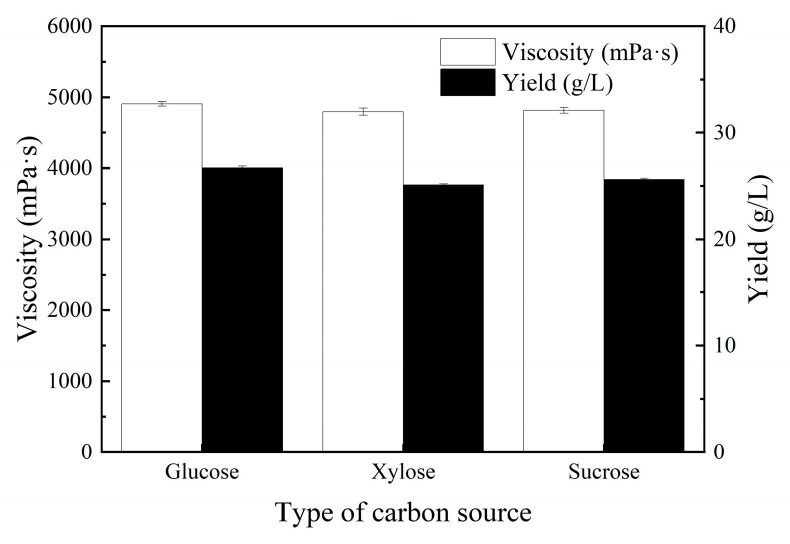
The effects of different carbon sources on the production of MT gum. The apparent viscosity was measured at 60 rpm with sp-4. Values were three replicates.

**Figure 3 polymers-17-00186-f003:**
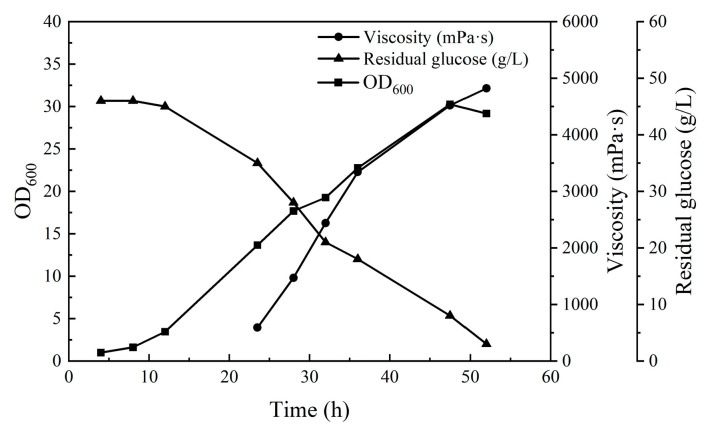
MT gum fermentation process under optimal production conditions.

**Figure 4 polymers-17-00186-f004:**
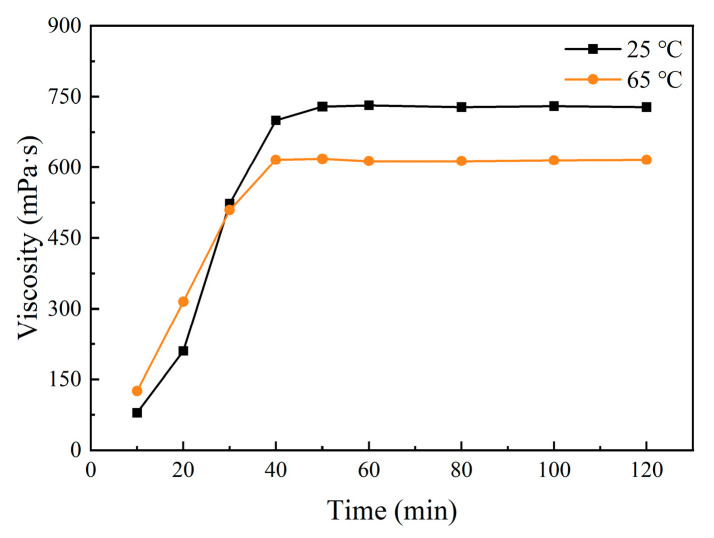
The dissolution time of MT gum at 5 g/L. The apparent viscosity was measured at 30 rpm with sp-2.

**Figure 5 polymers-17-00186-f005:**
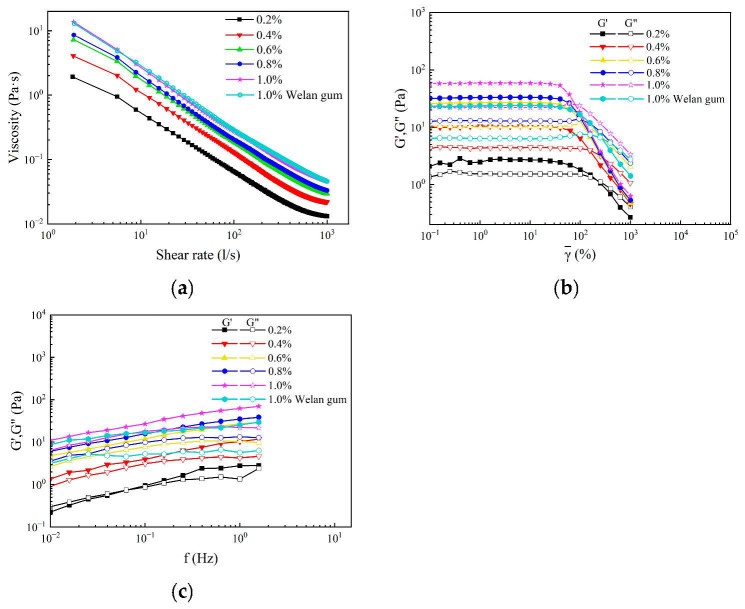
Rheological property measurements of MT gum. (**a**) The apparent viscosity of MT gum as a function of the shear rate. (**b**) The linear dynamic viscoelastic properties of the MT solutions. (**c**) Storage modulus (G′) and loss modulus (G”) as a function of frequency for MT gum.

**Figure 6 polymers-17-00186-f006:**
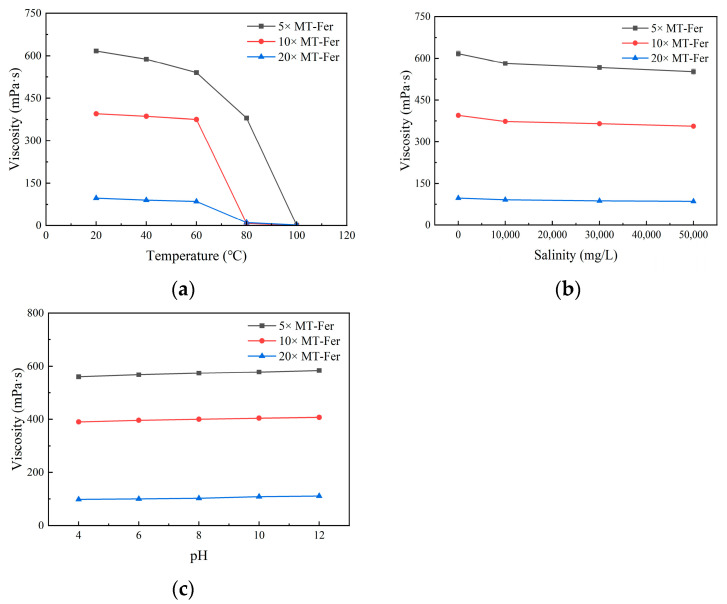
Effects of different factors on the apparent viscosity of MT-Fer solutions. (**a**) Temperature, (**b**) salinity, (**c**) pH. The viscosity was measured at 30 rpm with sp-2 (5×, 10×) and 50 rpm with sp-1 (20×). Values were three replicates.

**Figure 7 polymers-17-00186-f007:**
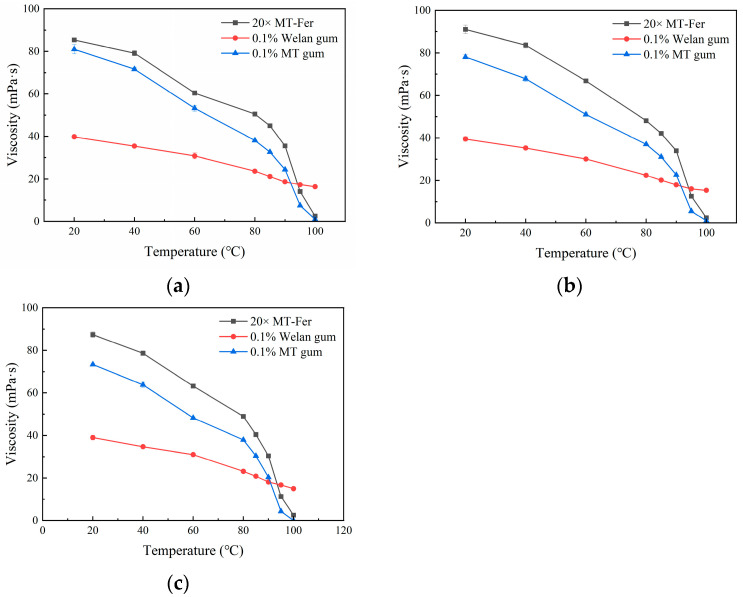
The apparent viscosity of MT gum and welan gum under temperature-salinity composite conditions. (**a**) 10,000 mg/L salinity; (**b**) 30,000 mg/L salinity; (**c**) 50,000 mg/L salinity. The apparent viscosity was measured at 50 rpm with sp-1. Values were three replicates.

**Figure 8 polymers-17-00186-f008:**
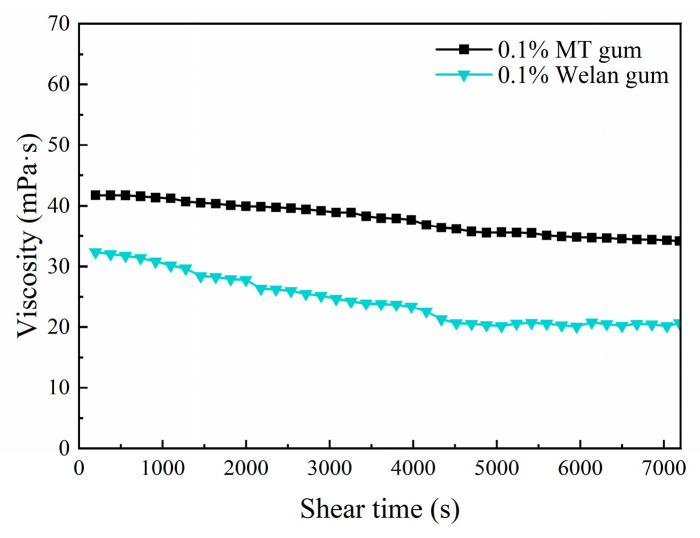
The changes in apparent viscosity of MT and welan solutions (0.1%, *w*/*v*) with shear time at 7.34 s^−1^ under conditions of 80 °C and 50,000 mg/L salinity.

**Figure 9 polymers-17-00186-f009:**
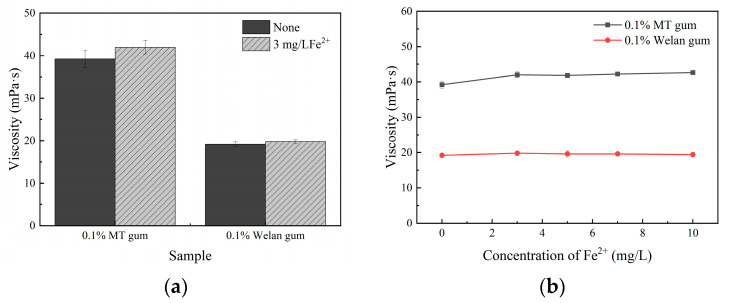
The effects of Fe^2+^ on the apparent viscosity of MT gum (0.1%, *w*/*v*) and welan gum (0.1%, *w*/*v*). (**a**) 3 mg/L Fe^2+^, (**b**) Different concentrations of Fe^2+^ (0, 3, 5, 7, 10, mg/L). The MT gum and welan gum were dissolved in brine water (50,000 mg/L salinity) with or without FeSO_4_. All solutions were incubated at 80 °C for 10 h, and then the viscosity was measured at 50 rpm with sp-1. Values were three replicates.

**Table 1 polymers-17-00186-t001:** Composition of the brine water.

Composition (mg/L)			Total Salinity (mg/L)
NaCl	CaCl_2_	MgCl_2_	
7582.9	900.5	1516.6	10,000
22,748.7	2701.5	4549.8	30,000
37,914.5	4502.5	7583	50,000

**Table 2 polymers-17-00186-t002:** The effects of temperature on the production of MT gum.

Temperature (℃)	Time (h)	Viscosity (mPa·s)	OD_600_	Yield (g/L)	Residual Glucose (g/L)
30	56	4760 ± 50 ^a^	32.26 ± 0.18 ^c^	25.3 ± 0.2 ^ab^	3 ± 0.1 ^b^
32	55	4720 ± 20 ^a^	33.15 ± 0.08 ^a^	25.1 ± 0.3 ^ab^	3 ± 0.2 ^b^
35	52	4800 ± 30 ^a^	32.97 ± 0.14 ^ab^	25.6 ± 0.3 ^a^	2 ± 0.1 ^c^
37	60	4530 ± 60 ^b^	32.61 ± 0.18 ^bc^	24.6 ± 0.1 ^b^	1 ± 0.1 ^d^
40	72	0 ^c^	3.06 ± 0.02 ^d^	0 ^d^	34 ± 0.5 ^a^

The apparent viscosity was measured at 60 rpm with sp-4. Values were the means ± LSD of three replicates. Different letters indicate significant differences at *p* < 0.05.

**Table 3 polymers-17-00186-t003:** Monosaccharide composition and molecular weight of MT gum.

Parameters	MT Gum
Monosaccharide composition (%)	
D-Galactose	0.77
D-Glucose	65.91
D-Mannose	1.57
L-Fucose	1.06
L-Guluronic acid	30.69
Molar mass moments (g/mol)	
Mw ^a^	7.24 × 10^5^
Mn ^b^	2.70 × 10^5^
PDI ^c^	2.68

^a^ Mw is the weight-average molecular weight. ^b^ Mn is the number-average molecular weight. ^c^ PDI is the polydispersity index.

**Table 4 polymers-17-00186-t004:** Production of different sphingans.

Sphingans	Srain	Temperature (°C)	Time (h)	Carbon Source	Yield (g/L)	References
MT	*Sphingomonas* sp. MT01	30–37	52	Glucose	25.6 ± 0.3	This work
Gellan	*Sphingomonas* sp. SM2	30	96	Glucose	8.64 ± 0.34	[46]
	*Sphingomonas paucimobilis* QHZJUJW CGMCC2428	30	112	Sucrose	19.90 ± 0.68	[47]
Welan	*Sphingomonas* sp. ATCC 31555	30	72	Sucrose	21.34 ± 0.26	[48]
	*Sphingomonas* sp. HT-1	37	66	Glucose	22.68 ± 0.50	[45]
Sanxan	*Sphingomonas sanxanigenens* NX02	30	84	Glucose	14.88 ± 0.83	[49]
WL	*Sphingomonas* sp. WG	30	90	Glucose	33.3	[50]

**Table 5 polymers-17-00186-t005:** The monosaccharide composition of different sphingans.

Sphingans	Monosaccharide Composition (%)							References
	D-Glucose	D-Mannose	L-Rhamnose	D-Galactose	L-Fucose	D-Glucuronic Acid	L-Guluronic Acid	
MT	65.91	1.57	-	0.77	1.06	-	30.69	This work
Welan	47.31	12.37	24.92	-	-	15.04	-	[51]
Gellan	60	-	20	-	-	20	-	[5]
Duitan	+	-	+	-	-	+	-	[6]
Sanxan	52.95	35.29	5.88	-	-	5.88	-	[35]
WL	42.22	18.52	39.26	-	-	-	-	[50]

“+” indicates the presence of this component; “-” indicates the absence of this component.

## Data Availability

Data are contained within the article and Appendix A.

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
