# Peer review of "A Novel Exopolysaccharide Produced by Sphingomonas sp. MT01 and Its Potential Application in Enhanced Oil Recovery"

_polymers, 2025, doi:10.3390/polym17020186_

Round 1

Reviewer 1 Report (Previous Reviewer 1)

Comments and Suggestions for Authors

The comments have been addressed.

Author Response

Thank you very much for taking the time to review this manuscript.

Reviewer 2 Report (Previous Reviewer 2)

Comments and Suggestions for Authors

The manuscript (polymers-3402303) titled "A novel exopolysaccharide produced by Sphingomonas sp. MT01 and its potential application in enhanced oil recovery" presents a promising investigation into the production and potential application of a novel biopolymer, MT gum, in enhanced oil recovery (EOR) within challenging double-high reservoir conditions. While the study offers valuable insights, a major revision is necessary to enhance its clarity, scientific rigor, and completeness. Below are the specific suggestions for improvement:

1. Line 12-22: Add specific quantitative results (e.g., viscosity retention, salinity tolerance) to strengthen the abstract.

2. Line 27-29: Clarify "unique structure and properties" with examples of sphingan applications.

3. Line 31-33: Highlight structural differences of sphingan compared to other biopolymers.

4. Line 64-69: Clearly state the research gap this study addresses. 5. Line 72-75: Justify why tropical plant rhizosphere soils were chosen for strain isolation.

6. Line 93-95: Explain the selection of fermentation conditions (e.g., agitation speed, aeration rate).

7. Line 137-139: Provide rationale for the mobile phase composition in HPLC analysis.

8. Line 148-153: Justify the use of a specific column for molecular weight determination.

9. Line 207-211: Clarify the basis for classifying MT01 as a novel strain. 10. Line 229-233: Add statistical significance for differences in fermentation conditions (Table 2).

11. Line 259-264: Emphasize the practical advantage of rapid MT gum dissolution for offshore use.

12. Figure 2 (Line 226): Label axes more descriptively (e.g., "Carbon Source Type," "Viscosity (mPa·s)").

13. Figure 5a (Line 273): Use clearer markers for different concentrations. 14. Table 3 (Line 254): Explain the relevance of monosaccharides for EOR applications in the caption.

15. Line 308-310: Elaborate on how hydrogen bonding contributes to stability.

16. Line 343-347: Explain why MT gum showed better viscosity under composite conditions than welan gum.

17. Line 379-380: Clarify the importance of Fe²⁺ resistance for offshore reservoirs.

18. Line 403-405: Discuss how shorter fermentation time and temperature tolerance support scalability.

19. Line 457-463: Include quantitative results (e.g., viscosity, salinity tolerance) in the conclusion.

Comments on the Quality of English Language

Throughout the manuscript: Ensure consistent terminology and concise phrasing, e.g., replace "high temperature and high salinity" with "double-high conditions."

Round 2

Reviewer 2 Report (Previous Reviewer 2)

Comments and Suggestions for Authors

The authors have responded adequately to all of my comments. The manuscript is now suitable for publication.

This manuscript is a resubmission of an earlier submission. The following is a list of the peer review reports and author responses from that submission.

Round 1

Reviewer 1 Report

Comments and Suggestions for Authors

Ø  The introduction section is very concise and interesting to read. However, the following improvements may enhance its quality:

§  Polymer use in enhanced oil recovery is mentioned as a “new field” in line 36. However, in China, polymer flooding contributed to enhanced oil recovery since 2004 which seems contradictory so some recent examples may be quoted here.

§  HPAM synthetic polymer is unstable at high temperatures and salt environment which is mentioned in line 46. If the reference paper had specific temperature and salt content limits HPAM withstands, those digits should be there which enhances the comparability with produced new polymers and more justify the rationale to develop new polymers.

§  For a new reader, the factual range of temperature and salt content within the double-high reservoir should be added with suitable citation/s.

§  The points of environmental pollution by synthetic or chemical-based polymers over biological polymers for a sustainable and pollution-free environment can be added. The methodology sections are well illustrated. The following suggestions might provide more clarity:

Ø  In the method section, the exploration of plant rhizosphere for isolation of EPS higher producers whose EPS has desired high salt and high-temperature stability is surprising. So, in place of an extremophilic habitat, why plant rhizosphere and which plants are utilized for the same should be added for readers’ interest.

§  The culture growth conditions viz. temperature and carbon sources were optimized for fermentation. However, in section 2.2 along with media composition, the applied temperature range should be included.

§  The yield of produced EPS was measured as g/L which is ok. However, to justify the stain as a higher producer weight (dry cell mass) by weight (gm EPS produced) means gm EPS per gm of cell mass should be measured.

§  The method of drying (temperature and other conditions) of EPS as crude and after purification (freeze drying) should be included in the 2.4 and 2.5.1 sections.

§  In the 2.5.2 section EPS derivatization or HPLC analysis is not cited with any reference, the methodology is developed by the author or referred from literature and modified or further optimized? Give justification for more clarity. Cite the article if any used to develop the composition check protocol.

§  The MT-fer was used to check the stability of EPS. However, purified EPS must be used for the same because sometimes the deproteinization via TCA or sevag reagents following chromatography destabilizes the EPS. If you did the purification of EPS, why EPS-fer was used to check stability? How does the stability of MT-fer justify the stability of purified MT?

Ø  In the result sections, the subsections should be in continuation with the method section for ease of reading.

§  In Tables and Figures, the units viz. mean ± SD/SE should be there in the footnote.

§  The viscoelasticity of MT polymer is better or comparable to the control polymer. Please justify.

§  For desired oil recovery, the EPS should not have emulsification action which is the property of a wide range of microbial EPSs. Give comments on it.

§  In 3.7.2, it was written that at low concentrations MT-fer revealed high stability. Provide the reasons for such an outcome.

§  In 3.7.3, the effect of pH is given. At acidic pH of 2-3 loss of viscosity resulted whose justification or reason should be discussed.

§  HPLC chromatogram of the sample and standard mix may enhance the quality of the result section.

§  Section 3.9, the effect of Fe ion on viscosity is informative.

Ø  The discussion, conclusion and abstract sections seem adequate.

Reviewer 2 Report

Comments and Suggestions for Authors

The manuscript (polymers-3369007) "A novel exopolysaccharide produced by Sphingomonas sp. MT01 and its potential application in enhanced oil recovery" explores the production and application of MT gum. However, the study suffers from significant shortcomings that limit its contribution to the field:

  1. Lack of Novelty: Many studies have already reported the production and application of exopolysaccharides like welan gum in enhanced oil recovery (EOR) https://doi.org/10.1007/s10924-023-03132-1 and https://doi.org/10.1016/j.ijbiomac.2024.130193. The manuscript does not adequately demonstrate how MT gum provides distinct advantages over existing biopolymers.
  2. Low Efficiency and Production: The reported yield and efficiency of MT gum production are relatively low compared to other sphingans https://doi.org/10.1016/j.ijbiomac.2020.08.114, which raises concerns about its scalability and practical utility in EOR.
  3. Routine Approach: The methods and experiments employed in the study are quite routine and lack innovative elements. The manuscript does not present any groundbreaking advancements in biopolymer production or application. https://doi.org/10.1038/srep37899

Hence, while the manuscript touches on an important topic, its lack of novelty, limited production efficiency, and routine methodology do not meet the standards for publication. The authors are encouraged to address these issues and develop a more impactful study before resubmission.

Comments on the Quality of English Language

N/A